# The role of tail dependence in estimating posterior expectations

## Abstract

Many tasks in modern probabilistic machine learning and statistics require estimating expectations over posterior distributions. While many algorithms have been developed to approximate these expectations, reliably assessing their performance in practice, in absence of ground truth, remains a significant challenge. In this work, we observe that the well-known $k$-hat diagnostic for importance sampling (IS) [1] can be unreliable, as it fails to account for the fact that the common self-normalized IS (SNIS) estimator is a ratio. First, we demonstrate that examining separate $k$-hat statistics for the numerator and denominator can be insufficient. Then, we we propose a new statistic that accounts for the dependence between the estimators in the ratio. In particular, we find that the concept of tail dependence between numerator and denominator weights contains essential information for determining effective performance of the SNIS estimator.

## 1 Introduction and background

Algorithms for Bayesian computation continue to be used for increasingly complex probabilistic models, remaining an active research field [2]. Yet, in the absence of ground truth, it remains challenging in practice to determine how and in which sense an approximate inference algorithm has found a "good" solution, as studied by several recent works, for Markov Chain Monte Carlo (MCMC) [3–5], variational inference (VI) [6–8], and importance sampling [1, 9–11] (the latter two being closely connected). In this work, we focus on diagnostics that apply to IS and VI algorithms.

**Problem statement.** Let $\theta \in \Theta$ (commonly, $\mathbb{R}^{d_\theta}$) be the parameter of a Bayesian statistical model $\{p(y|\theta)\}_\theta$ for data $y \in \mathcal{Y}$ with posterior PDF $\pi(\theta|\mathcal{D}) \stackrel{\text{def}}{=} Z_\pi^{-1} \cdot \widetilde{\pi}(\theta|\mathcal{D}) = Z_\pi^{-1} \cdot \prod_n p(y_n|\theta) \cdot \pi(\theta)$ with $\mathcal{D} \stackrel{\text{def}}{=} \{y_n\}_{n=1}^N$, $Z_\pi$ the normalizer and prior PDF $\pi(\theta)$. Formally, we aim at constructing Monte Carlo estimates of a posterior expectation $I \in \mathbb{R}_{>0}$, defined as

$$I \stackrel{\text{def}}{=} \mathbb{E}_{\pi(\theta|\mathcal{D})}[f(\theta)] = \int f(\theta)\pi(\theta|\mathcal{D})d\theta, \tag{1}$$

where $f : \Theta \to \mathbb{R}_{\geq 0}$ is a suitably integrable test function. In particular, we are interested in obtaining diagnostics to determine the quality of an estimator $\widehat{I}$. As a concrete example, when we set $f(\theta) = p(y^{(n+1)}|\theta)$ for a test point $y^{(n+1)}$, $I$ is often written as $p(y^{(n+1)}|\mathcal{D})$, i.e., the evaluation of the posterior predictive PDF $p(y|\mathcal{D})$ at point $y^{(n+1)}$.[1]

**Self-normalized IS, combination with VI.** Approximating integrals like in Eq. (1) accurately is challenging. MCMC is a natural solution, but there are notable cases where it is not appropriate. For

---

[1]Such integrals can be used for estimating the predictive performance of a posterior [12] or the influence of a particular observation.

Submitted to Workshop on Bayesian Decision-making and Uncertainty, 38th Conference on Neural Information Processing Systems (BDU at NeurIPS 2024). Do not distribute.

example, when even exact i.i.d. sampling from $\pi(\theta|\mathcal{D})$ is inefficient, or when it is too expensive. In these cases one usually resorts to IS [13], where we obtain samples from a chosen proposal PDF $q$, as $\theta^{(s)} \overset{\text{i.i.d.}}{\sim} q(\theta)$, and construct estimators for $I$ as

$$\widehat{I}_{\text{SNIS}} = \sum_{s=1}^{S} \overline{w}^{(s)} f(\theta^{(s)}) \qquad , \overline{w}^{(s)} \overset{\text{def}}{=} \frac{w^{(s)}}{\sum_{s'=1}^{S} w^{(s')}}, \ w^{(s)} = w(\theta^{(s)}) = \frac{\widetilde{\pi}(\theta^{(s)}|\mathcal{D})}{q(\theta^{(s)})}. \tag{2}$$

Many theoretical properties of this estimator are known (see, e.g., [14] for a review). When the normalizing constant $Z_\pi$ is unknown (i.e., almost always), the normalization of the weights in Eq. (2) is not optional. In practice, it is difficult to find a good proposal, i.e., leading to estimates that are close to $I$. It is natural to use proposals that are the result of a VI algorithm [6], which is done implicitly or explicitly in the VI literature. See [6, 15–25] as examples for the many connections between VI and IS. A consequence of using a bad proposal is that the distribution of the weights $w_s$ tends to have a few very large values.

**Pareto-smoothed IS.** Exploiting the above observation, [1] proposed Pareto-smoothed IS (PSIS), which replaces the largest $M$ unnormalized weights [2] to get SNIS estimators with better behaviour. They fit a generalized Pareto distribution (GPD) to the weights $\{w^{(s)}\}_{s=1}^{S}$. The new ("smoothed") weights introduce bias but reduce variance. The GPD has three parameters, the most important of which is the shape parameter $k$. [1] propose to use an estimate of $k$, i.e., $\widehat{k}$, as a diagnostic for IS.

**The $\widehat{k}$ diagnostic.** [1] use the estimated value of $k$, i.e., $\widehat{k}$, as a diagnostic for deciding whether the SNIS estimates with PSIS-corrected weights are reliable. The GPD has $1/k$ finite fractional moments when the true $k > 0$, which suggests finite variance as soon as $k < 0.5$. Note that this guarantees finite variance only for the normalizing constant estimator $\widehat{Z}_\pi = 1/S \sum_{s=1}^{S} w^{(s)}$, which is implicit in the denominator of SNIS [26]. [1] find empirically that when $S > 2000$, estimation with PSIS-corrected weights is reliable for $\widehat{k} < 0.7$, a threshold less stringent than $0.5$. An advantage of $\widehat{k}$ is that it is not an IS estimate itself, unlike the effective sample size (ESS) [10], attempting to address the issues with variance-based diagnostics [9].

## 2 Methodology

Several works [26–28] have shown theoretically and empirically that accurately estimating posterior expectations such as $I$ in Eq. (1) involves more than simply finding a proposal $q(\theta)$ that is close to the posterior $\pi(\theta|\mathcal{D})$. This is because the SNIS estimator is a ratio estimator, as $I$ itself is the ratio of two integrals,

$$I = \frac{\int f(\theta)\widetilde{\pi}(\theta|\mathcal{D})d\theta}{\int \widetilde{\pi}(\theta|\mathcal{D})d\theta} \overset{\text{def}}{=} \frac{I_{\text{num}}}{Z_\pi} \overset{\text{def}}{=} \frac{I_{\text{num}}}{I_{\text{den}}}, \tag{3}$$

where we relabelled the normalizing constant $I_{\text{den}}$. Therefore, we can write the SNIS estimator as

$$\widehat{I}_{\text{SNIS}} = \frac{\frac{1}{S} \sum_{s=1}^{S} w^{(s)} f(\theta^{(s)})}{\frac{1}{S} \sum_{s=1}^{S} w^{(s)}} = \frac{\widehat{I}_{\text{num}}}{\widehat{I}_{\text{den}}}, \ \theta^{(s)} \overset{\text{i.i.d.}}{\sim} q(\theta), \tag{4}$$

where the two estimators $\widehat{I}_{\text{num}}$ and $\widehat{I}_{\text{den}}$ are unbiased, but $\widehat{I}_{\text{SNIS}}$ is not. As elaborated in [26], the asymptotic variance of the SNIS estimator is driven by the variance of the numerator estimator, the variance of the denominator, and the covariance between them. For convenience, we define two unnormalized importance weight functions, the one used in the numerator for $\widehat{I}_{\text{num}}$ and the one used in $\widehat{I}_{\text{den}}$, as

$$w_{\text{num}}(\theta) = \frac{f(\theta)\widetilde{\pi}(\theta|\mathcal{D})}{q(\theta)}, \qquad w_{\text{den}}(\theta) = \frac{\widetilde{\pi}(\theta|\mathcal{D})}{q(\theta)}. \tag{5}$$

We can then write the SNIS estimator as a ratio of two unbiased IS estimators,

$$\widehat{I}_{\text{SNIS}} = \frac{\frac{1}{S} \sum_{s=1}^{S} w_{\text{num}}(\theta^{(s)})}{\frac{1}{S} \sum_{s=1}^{S} w_{\text{den}}(\theta^{(s)})}, \ \theta^{(s)} \overset{\text{i.i.d.}}{\sim} q(\theta). \tag{6}$$

---

[2]See [1] for the choice of $M$.

Given that there are two IS weights, $w_{\text{num}}(\theta^{(s)}), w_{\text{den}}(\theta^{(s)})$ in the above, it is natural to consider that one may track reliability $\widehat{I}_{\text{SNIS}}$ by computing two diagnostics $\widehat{k}_{\text{num}}, \widehat{k}_{\text{den}}$ separately for weights $\{w_{\text{num}}^{(s)}\}_{s=1}^{S}$ and $\{w_{\text{den}}^{(s)}\}_{s=1}^{S}$. [1] explored this option empirically, reporting that in their experiments it was sufficient to take $\max(\widehat{k}_{\text{num}}, \widehat{k}_{\text{den}})$ to determine reliability of the ratio. In this work, we will argue that this heuristic misses useful information and propose a new diagnostic.

## 2.1 Capturing error cancellation with tail dependence

The diagnostics $\widehat{k}_{\text{num}}$ and $\widehat{k}_{\text{den}}$ describe how well $\widehat{I}_{\text{num}}$ and $\widehat{I}_{\text{den}}$ respectively approximate $I_{\text{num}}$ and $I_{\text{den}}$, serving as an (improved) substitute for estimates of variance (like the ESS). Yet, the variance of the SNIS estimator $\widehat{I}_{\text{SNIS}}$ is not only affected by the variance of the numerator of Eq. (6), the variance of the denominator. It is also affected by the covariance $\text{Cov}_q[\widehat{I}_{\text{num}}, \widehat{I}_{\text{den}}]$ [26].

A straightforward idea to capture this missing piece of information from $\widehat{k}_{\text{num}}$ and $\widehat{k}_{\text{den}}$ is to construct an estimate of $\text{Cov}_q[\widehat{I}_{\text{num}}, \widehat{I}_{\text{den}}]$, using the same samples from $q$ used to estimate $I$. Yet, doing so would suffer the same drawbacks of variance-based diagnostics, which was a motivation for $\widehat{k}$ [1]. Thus, we will develop a diagnostic that is not a direct estimate of $\text{Cov}_q[\widehat{I}_{\text{num}}, \widehat{I}_{\text{den}}]$. Like [1], we also exploit the fact that the distribution of $w_{\text{num}}$ and $w_{\text{den}}$ can be well approximated with a power-law distribution in the tails. Specifically, we will look at a suitable notion of dependence between the tails of $w_{\text{num}}$ and $w_{\text{den}}$. This notion will replace the covariance $\text{Cov}_q[\widehat{I}_{\text{num}}, \widehat{I}_{\text{den}}]$ as our target estimate. In fact, covariance, up to normalization, is equivalent to Pearson's correlation $\rho$, which is only a very specific form of dependence, with many known limitations [29].

**Dependence and error cancellation.** An intuition for why higher covariance between the estimators $\text{Cov}_q[\widehat{I}_{\text{num}}, \widehat{I}_{\text{den}}]$, or other dependence metrics, can lead to lower error is that, in a ratio, error cancellation can happen. Error cancellation in ratios has been exploited to derive better convergence rates for other numerical integration methods [30]. In IS, it is known that large IS weights lead to high errors. Therefore, error cancellation in the ratio of Eq. (6) could happen when a large weight in the numerator is offset by another similarly large weight in the denominator. We now formalize this using the notion of tail dependence.

**Definition 1 (Upper tail dependence coefficient and tail dependence)** *Let $W_1, W_2$ be two real-valued random variables. Let their (continuous) marginal CDFs be $F_1, F_2$. Then,*

$$\lim_{q \to 1^-} \mathbb{P}\left[W_2 > F_2^{-1}(q) | W_1 > F_1^{-1}(q)\right] = \lambda_U, \tag{7}$$

*provided the limit exists, is known as upper tail dependence coefficient $\lambda_U \in [0,1]$. If $\lambda_U > 0$, we say that $W_1, W_2$ are asymptotically tail dependent, with the magnitude of $\lambda_U$ determining the strength of depedence.*

Next, we discuss how to relate the above concept to the estimation of $I$.

## 2.2 Proposed reliability checks

We propose to diagnose whether the estimate in Eq. (6) is reliable by examining three quantities: $\widehat{k}_{\text{num}}$, $\widehat{k}_{\text{den}}$ and a new diagnostic that is constructed as an approximation of the tail dependence coefficient $\lambda_U$ between $w_{\text{num}}, w_{\text{den}}$. Our aim is to study how these quantities relate to the effective performance of $\widehat{I}_{\text{SNIS}}$ as an estimator of $I$, which we define as follows.

**Definition 2 (Effective performance)** *We define the effective performance of an estimator $\widehat{I}$ of $I$ as ensuring that the value of $(\widehat{I}/I)$ is close to 1 with high probability. This takes into account the possibility of $I$ being very small, e.g., $10^{-7}$ following the reccomendation of [9]. In log-space, it is equivalent to look at how $\log I - \log \widehat{I}$ is close to zero (recall $I > 0$).*

**Semi-parametric estimation of tail dependence** In mathematical finance, various estimators of tail dependence have been developed [31–33]. We begin by studying semi-parametric estimators, following the assumption used by [1] and common in heavy-tailed distribution inference [34].

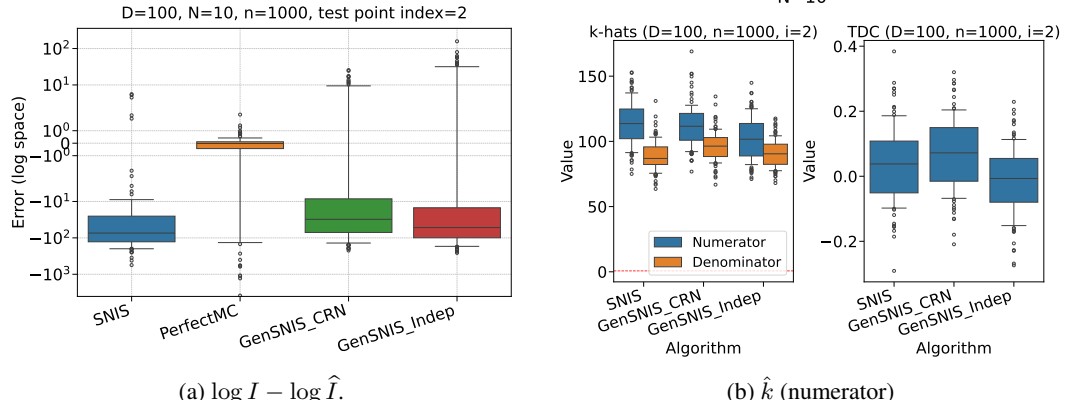

(a) $\log I - \log \widehat{I}$.

(b) $\widehat{k}$ (numerator)

Figure 1: Results ($d_\theta = 100$) over 100 replications. We compare SNIS, GenSNIS (see Section 3) with a common random number (CRN) and GenSNIS with independent marginals. From Fig. 1a, we see that GenSNIS with CRN performs best; this cannot be captured by $\widehat{k}$ values, but only by the higher TDC. Note that in such high dimension all methods perform poorly. We found similar results for lower dimensions and showcase here only a high-dimensional case.

109 Specifically, we assume the distribution of $w_{\text{num}}, w_{\text{den}}$ is well approximated by a GPD in the tails.
110 Similarly, to estimate tail dependence, we assume the *copula* of their joint distribution is well ap-
111 proximated by an extreme value copula [35], also only in the tails.[3] We hypothesize that tail depen-
112 dence between $w_{\text{num}}$ and $w_{\text{den}}$ improves $\widehat{I}_{\text{SNIS}}$ performance, similar to the effect of $\text{Cov}_q[w_{\text{num}}, w_{\text{den}}]$,
113 but easier to estimate and more reliable.

## 3   Preliminary results on Bayesian linear regression and conclusions

115 We look at the distribution of $\log I - \log \widehat{I}$ over different replications. We consider estimating the
116 posterior predictive of a Bayesian linear regression (BLR) model where we can compute the exact
117 value of $I$. That is, from Eq. (1), we set $f(\theta) = p(y^{(n+1)}|\theta)$ for a test point $y^{(n+1)}$, and $\pi(\theta|\mathcal{D})$ is a
118 Gaussian with known mean and covariance (BLR posterior).[4]

119 To validate our hypothesis that tail dependence contains useful information, we check the behaviour
120 of the diagnostics $\widehat{k}_{\text{num}}, \widehat{k}_{\text{den}}$ our tail dependence diagnostic $\widehat{\lambda}_U$ estimated from a *Gumbel copula*
121 $C(u_1, u_2; \rho, \theta)$ (which we found performing better than a t-copula), given by $2 - 2^{1/\theta}$.[5] We find
122 that, when $k$-diagnostics between competitors are similar for numerator and denominator, a higher
123 tail dependence coefficient (TDC) explains the better performance. To explain our results, we need
124 to introduce a recent generalization of the SNIS estimator proposed in [26], i.e., sampling from
125 an extended space $\mathbb{R}^{d_\theta} \times \mathbb{R}^{d_\theta}$, as $\widehat{I}_{\text{GenSNIS}} = \frac{\frac{1}{S}\sum_{s=1}^{S} w_{\text{num}}(\theta_1^{(s)})}{\frac{1}{S}\sum_{s=1}^{S} w_{\text{den}}(\theta_2^{(s)})}, \ [\theta_1^{(s)}, \theta_2^{(s)}] \overset{\text{i.i.d.}}{\sim} q_{1,2}(\theta_1, \theta_2)$. SNIS
126 is a special case where the joint is a degenerate joint with $\theta_1 = \theta_2$. Another special case is tak-
127 ing $q_{1,2}(\theta_1, \theta_2) = q_1(\theta_1)q_2(\theta_2)$, which is done in previous works including notably target-aware
128 Bayesian inference [27]. Finally, for these experiments we consider the choice of $q_{1,2}(\theta_1, \theta_2)$ that
129 uses a common random number (CRN) for numerator and denominator, but has different marginals.
130 Concretely we used Gaussian proposals $\mathcal{N}(\theta_1; \mu_1, \Sigma_1)$ and $\mathcal{N}(\theta_2; \mu_2, \Sigma_2)$ for numerator and de-
131 nominator, respectively. The parameters are set to the optimal ones (given by the BLR true posteri-
132 ors for numerator and denominator) perturbed by an error term. The SNIS estimator uses only one
133 distribution $q(\theta)$, so we take the midpoint between the two optimal IS means and covariances for its
134 parameters. Fig. 1 shows the results. We indeed find in other settings (for $d_\theta$, noise variance, and co-
135 variate distributions) that when $\widehat{k}$ values are similar for numerator and denominator, tail dependence
136 explains the remaining performance if a difference exists. We plan to test further TDC metrics and
137 Bayesian models.

---

[3] A copula of a bivariate joint distribution is the distribution on $[0, 1]^2$ after transforming the marginals to the uniform distribution. Many parametric copula families exist [36].

[4] See [37] for expressions about BLR including closed form posterior predictives.

[5] We use the estimate of $\widehat{\rho}$ from the Python statsmodels package, while setting $\nu$ manually.

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
