# OpenReview forum: "The role of tail dependence in estimating posterior expectations"
_NeurIPS.cc/2024/Workshop/BDU — NeurIPS BDU Workshop 2024 Poster_

### Official Review · Reviewer_eTFq · 2024-09-23
**An interesting paper and approach but some parts of it are not clear and the experiments are not thorough enough.**

**Rating:** 6
**Confidence:** 3

**Review:**

Pros:
- The paper introduces an interesting hypothesis - tracking tail dependence between estimators in the ratio of the SNIS estimator can capture the information lost in the covariance between them.

Cons:
- As the authors wrote, more experiments should be done to showcase the importance of tail dependence.
- Perhaps I didn't understand something, but I found some parts not clear enough. In particular,
  - The explanation for not estimating $Cov[\hat{I_{num}}, \hat{I_{den}}]$ in lines 78-80.
  - The explanations about the experimental part (lines 123-138) were too succinct. Perhaps the authors could elaborate more in the appendix (e.g., student-t copula, sampling from an extended space)
- Minor:
  - "..we we.." in line 9
  - $w_s$ instead of $w^{(s)}$ in line 39

Questions:
- How does the diagnostic behave with larger dimensionality? Does it have an effect on it?

---

### Official Review · Reviewer_uqJK · 2024-10-07
**Improved SNIS diagnoistics by assuming tail dependence**

**Rating:** 6
**Confidence:** 3

**Review:**

The author proposed using tail dependence coefficents to improve the current SNIS diagnostics in estimating posterier expectation. The proposed TDC metrics highlighted important posterior diagnostic insights not availlable with the k-hat statistics. The paper generally flows well and is nicely presented overall, in particular section 1 is well-written with detailed mathematical background.

However, the reviewer believes, in its current form, the paper has still several places to work on:
* elaborate the concept of tail dependence in greater details in Definition 1 and Equation 7
* provide more reasoning in the assumptions from line 112-114, to what extend is this applicable in fields beyond mathmatical finance?
* clearly label the y-axis in Figure 1 b), c) and d), are we still in log space?
* one generally expects the experimental results and conclusion are written in separates sections, given there is only one experiment being done, the current merge of the two is understandable.
* still, the case study presented with BLR is perhaps too simple, as the reviewer believes in such case MCMC estimates of the posterior will not be too expensive to compute. It will be better if the author can think if additional case study to better demonstrate the advantages proposed by using the TDC metrics

---

### Official Review · Reviewer_jkmk · 2024-10-09
**Accep,**

**Rating:** 7
**Confidence:** 3

**Review:**

The demo results are somewhat underwhelming, and while the paper covers a lot of ground, the demo itself feels overly simplistic. There is also a lack of comprehensive tests to assess the control of tail dependence, and the mathematical proofs (e.g., provability of upper and lower bounds) are missing.

That said, the writing is solid, and the paper is well-formatted. I believe it offers a promising direction, and I hope the authors can expand on the demos and provide new insights for the community.

---

### Decision · Program_Chairs · 2024-10-09

Accept (Poster)